# Emerging Opportunities in Human Pluripotent Stem-Cells Based Assays to Explore the Diversity of Botulinum Neurotoxins as Future Therapeutics

**DOI:** 10.3390/ijms22147524

**Published:** 2021-07-14

**Authors:** Juliette Duchesne de Lamotte, Anselme Perrier, Cécile Martinat, Camille Nicoleau

**Affiliations:** 1IPSEN Innovation, 91940 Les Ulis, France; juliettedelamotte@wanadoo.fr; 2I-STEM, INSERM UMR861, Université Evry-Paris Saclay, 91100 Corbeil-Essonne, France; 3Laboratoire des Maladies Neurodégénératives: Mécanismes, Thérapies, Imagerie, CEA/CNRS UMR9199, Université Paris Saclay, 92265 Fontenay-aux-Roses, France

**Keywords:** botulinum neurotoxins, cell-based assays, human pluripotent stem cells, high-throughput, BoNT-based therapeutics

## Abstract

Botulinum neurotoxins (BoNTs) are produced by Clostridium botulinum and are responsible for botulism, a fatal disorder of the nervous system mostly induced by food poisoning. Despite being one of the most potent families of poisonous substances, BoNTs are used for both aesthetic and therapeutic indications from cosmetic reduction of wrinkles to treatment of movement disorders. The increasing understanding of the biology of BoNTs and the availability of distinct toxin serotypes and subtypes offer the prospect of expanding the range of indications for these toxins. Engineering of BoNTs is considered to provide a new avenue for improving safety and clinical benefit from these neurotoxins. Robust, high-throughput, and cost-effective assays for BoNTs activity, yet highly relevant to the human physiology, have become indispensable for a successful translation of engineered BoNTs to the clinic. This review presents an emerging family of cell-based assays that take advantage of newly developed human pluripotent stem cells and neuronal function analyses technologies.

## 1. Introduction

Botulism is a potentially fatal disorder of the nervous system affecting both humans and animals, most often after ingestion of food contaminated with bacteria or spores. Botulism results in progressive flaccid paralysis of motor and autonomic nerves. It was first described in the 1820s by Kerner, who presented a study on several patients suffering from fatal poisoning after ingestion of contaminated sausages. Kerner described in these patients muscular paralysis of respiratory muscles, muscles of the upper and lower limbs, and vegetative disorders such as mydriasis, double vision, and gastrointestinal and bladder disorders [1,2,3]. “Kerner’s disease” as it was originally called, later became “botulism” (from the Latin botulus, sausage). The number of cases of foodborne botulism increased around the world during the 19th century, mainly as a result of the consumption of sausages but also due to smoked fish and low-acid preserved vegetables.

In 1895, Ermengem isolated for the first time an anaerobic bacterium, called Clostridium botulinum, from a contaminated ham as well as the intestine and spleen of botulinic patients [4,5]. After the discovery of a new strain of toxin serologically distinct from the one isolated by Ermengem, Burke later established in 1919 an alphabetical nomenclature for the botulinum neurotoxin postulating the existence of different species of Clostridium botulinum producing serologically different toxins, botulinum neurotoxins A and B (BoNT/A and BoNT/B) [6,7,8,9]. In the 1920s, Sommer and Snipe isolated the neurotoxin for the first time, which lead twenty years later to the development of a purification process of the neurotoxin protein in its crystalline form by Lamanna and colleagues [10,11].

Despite being considered as one of the most poisonous substance known, with a lethal dose estimated at 0.1–1 ng/Kg in intravenous and 1 µg/Kg orally in Humans, botulinum neurotoxins (BoNTs) have, over the years, successfully become a therapeutic agent for a wide spectrum of disorders especially neurological disorders when injected locally, as well as for aesthetic applications [12,13,14,15,16,17]. Initially used to treat strabismus, the increasing understanding of the biology of the neurotoxins and the availability of distinct toxin serotypes and subtypes offers the prospect of expanding BoNTs indications and resulting therapeutic benefit to a greater range of clinical conditions. One of the major roadblocks facing pharmaceutical development of BoNTs remains the access to robust, affordable humanized cell-based assays. Innovative approaches in the areas of humanized cell-based assays, combined with highly sensitive analytical tools, will be key for the successful development of next generation BoNTs drugs.

The recent advances in stem cell biology have raised expectations for BoNTs research, especially in light of the recent developments in human pluripotent stem cells technologies. Human pluripotent stem cells (hPSCs) are characterized by two main cardinal properties: a capacity to give rise to all the cell type forming an organism, and the capacity to self-renew almost without any limitation. When combined together, both properties of hPSCs offer, in theory, unlimited access to highly relevant cell source for a range of applications in drug discovery. Two sources of hPSCs, embryonic stem cells (ESCs) and induced pluripotent stem cells (iPSCs), have shown the most promise in the fields of regenerative and transplant medicine, disease modeling, high-throughput screens for drug discovery and development, and human developmental biology [18].

In this review, we examine recent advances in cellular models for the study of BoNTs most particularly those based on hPSCs. We describe their suitability for high-throughput studies including drug screenings aimed at identifying and evaluating novel BoNT-based therapeutics. The design of high-content stem cell-based models has the potential to extend our understanding of BoNTs intoxication process and help harness therapeutic potential of possibly new classes of compounds based on BoNTs. As the development of BoNT-based compounds is often marred by the lack of physiologically relevant and predictive assays, the use of stem cell-based models offers great opportunity as a valuable and a sensitive system to study biological effects of BoNT in a time and cost-efficient manner. Stem cell models can, under many conditions, provide an alternative to the animal models generally used in the field [19].

## 2. Botulinum Neurotoxins

In the last decades, a growing number of studies have contributed to describe the mechanisms by which the BoNTs block neuromuscular transmission [20,21], the genetic determinant of their activities, their structures, and mode of action at the molecular level [22,23,24,25].

BoNT in its native form is protected by multi-protein complexes composed of proteins associated with non-toxic neurotoxins, the Non-Toxic Non-Hemagglutinin (NTNHA) proteins [26,27]. These proteins support the transcytosis of BoNT across the intestinal barrier and protect BoNT from gastrointestinal degradation and release it into the circulation The BoNT thus released migrates to the neuromuscular junctions where it enters the neurons. The free BoNT is a 150 KDa polypeptide composed of two specific chains: an enzymatically active light chain (LC) fragment of 50 KDa attached to a heavy chain (HC) fragment of 100 KDa (Figure 1). Both chains are linked with an essential disulfide bridge and with a loop from the HC that wraps around the LC [28,29]. The HC is divided into two sub-domains including a C-terminal specific neuronal receptor binding domain and a translocation domain. The LC is a zinc endopeptidase that can cleave one or more specific proteins of the neuronal soluble N-ethylmaleimide-sensitive factor attachment protein receptor (SNARE) complex [24,25,30,31]. Neuronal SNARE proteins include syntaxin, synaptosomal-associated protein 25 KDa (SNAP25), and vesicle-associated membrane protein (VAMP, also called synaptobrevin). These SNARE proteins constitute the central components of the eukaryotic molecular machinery that mediate membrane fusion during trafficking and exocytosis of neurotransmitters at the axonal presynaptic terminal allowing the signal communication to the postsynaptic neuron. Neurotransmitters are stored in synaptic vesicles that fuse with the plasma membrane under the action of SNARE proteins in a calcium (Ca^2+^)-dependent manner. The cleavage of neuronal SNARE proteins by BoNT prevents membrane fusion and blocks the release of cholinergic neurotransmitters at the neuromuscular junction, ultimately leading to neuroparalysis.

Neutralizing antibodies can distinguish at least seven different serotypes of BoNT, from A to G, that are structurally similar but antigenically distinct. Several studies reported that monoclonal antibodies can neutralize only some but not all serotypes suggesting variability within each neurotoxin serotype. Such variabilities are now revealed and explained through sequencing of neurotoxin-encoding genes of bacteria infecting patient [32]. A BoNT protein that was tentatively named BoNT/H has been recently identified and corresponds to a chimeric toxin, with its LC similar to the LC of BoNT/F and its HC similar to the HC of BoNT/A. Furthermore, another apparently new BoNT serotype named BoNT/X, was recently identified [33,34]. BoNT/X has the lowest sequence identity with others serotypes and is not recognized by antisera against known (A–G) serotypes. All serotypes differ in their toxicity, molecular site of action, efficiency in terms of muscle paralysis, duration of effects, and specific affinity for their targets [34,35,36,37,38,39]. Each serotype has, for example, distinct binding and/or affinity specificity for different SNARE proteins (Table 1).

Physiologically, the neuromuscular transmission starts when a nerve impulse from peripheral or central nervous system reaches the nerve terminal of the neuromuscular junction (NMJ). At the synapse, the depolarization of the presynaptic membrane triggers the opening of voltage-gated calcium channels. This increase of intracellular free Ca^2+^ level triggers the membrane fusion activity of SNARE complexes, resulting in exocytosis of their content, the acetylcholine (ACh), into synaptic cleft. ACh then binds to its receptor on the surface of the muscle fiber triggering a new nerve impulse that spreads rapidly along the muscle fiber to make it contract [65,66]. Upon intoxication, BoNT moves to the NMJ, enters host neurons, and inhibits neurotransmitter release by a four-step process (Figure 2) [25,67,68,69,70,71].

The mechanism of action of BoNTs first involves their binding to receptors in the presynaptic membrane of nerve terminals. The active site of binding localized in the C-terminal HC portion of BoNTs, recognizes two distinct types of receptors simultaneously. The first type corresponds to gangliosides receptors abundantly expressed on neuronal plasma membrane and known to interact with BoNT with low affinity [72,73]. The second type are protein receptors, i.e., synaptic vesicle protein (SV2A, B, and C) and synaptotagmin (SYT-I and II). Upon binding to their SV2 or SYT receptors, BoNTs are co-internalized during endocytosis of these receptors in the context of their continuous recycling. After endocytosis, a conformational change of the BoNTs is triggered by endosome acidification. This conformational change allows the HC to form a channel and translocate the LC across the membrane into the cytoplasm. Once into the cytosol, BoNTs LC binds and cleaves transmembrane SNARE proteins on the membrane of cholinergic vesicles which consequently prevents the SNARE complex formation and leads to the inhibition of the release of ACh into the synaptic cleft. The LC of BoNT/A and BoNT/E specifically cleave synaptosomal-associated protein 25 KDa (SNAP25), while BoNT serotypes B, D, F, G, and H target synaptobrevin (VAMP1/2/3). BoNT/C cleaves both SNAP25 and syntaxin. BoNT/X cleaves VAMP1/2/3/4/5 and Ykt6.

The direct consequence of the inhibition of ACh release is a reduction in muscle contractions and efferent signaling at the NMJ, ultimately causing neuroparalysis. This inhibition of synaptic transmission is temporary and fully reversible, because a synaptic remodeling and a recovery process takes place at the NMJ which restores synaptic contacts and rescues functional neurotransmitter release [74,75,76]. Indeed, a sprouting phenomenon of terminal endings is observed at the NMJ after prolonged inhibition. This phenomenon of germination of axonal nerve endings gives rise to the regrowth of new synapses, which leads to the renewal of the pool of synaptic vesicles and contributes to functional recovery of the NMJ. The germination event then continues to expand long after neurotransmission is functional again [77]. Once the major nerve ending regains its maximum capacity to release neurotransmitters, the germination network loses its activity and is eliminated. Due to differences in receptor binding, speed of internalization and SNARE cleavage, and duration of effect, the various different BoNT serotypes available offer an interesting range of paralysis longevities.

BoNTs present significant opportunities to identify new ways to treat various clinical indications when used in low concentrations. Proper choice of dosage and site of administration are essential determinants of a positive beneficial effect to BoNTs treatment. Despite the great diversity of natural BoNTs, only serotypes A and B are commercially available so far. In addition to being approved by the US Food and Drug Administration (FDA) (Table 2), BoNTs have also been approved in many European countries by the European Medicines Agency (EMA) and by the UK Medicines and Healthcare Products Regulatory Agency (MHRA) for similar clinical indications. Some FDA-approved indications are still in the process of evaluation by EMA member countries [78,79].

The natural diversity of BoNTs as well as the increasing knowledge of their activities provide the opportunity to address novel clinical applications. In this context, the development of new cell-based assays to study biological effects of these neurotoxins may help to reduce the time and cost of current models. Several are undergoing investigations and clinical trials test the efficiency of BoNTs for new therapeutic indications, i.e., epilepsy [80], tremor [81], tics [82], depression [83], and endometriosis [84]. Up to now, the majority of the current assays used to characterize BoNTs are conducted in animal models, which may be limited by species-specificity affecting potential for translation to the clinic. The versality of hPSCs, in particular their capacity to generate neuronal cell types relevant to the study of BoNTs, position hPSCs as promising new players in the field.

## 3. BoNT Detection Assays

### 3.1. Rodent Bioassays

#### 3.1.1. In Vivo Bioassay

Currently, the standard assay to test the presence of BoNTs relies on the in vivo mouse bioassay (MBA) [85]. This assay is routinely used to detect toxin in suspect contaminated food and environmental samples of botulism and to assess the potency of therapeutic drug products [86,87]. It is based on the intraperitoneal or intravenous injection of suspected samples in mouse. Animals are then daily monitored for appearance of typical botulism symptoms such as muscle weakness and respiratory failure, which typically present within one to four days. This assay therefore presents the advantages of being able to detect biologically active toxins independently of their serotypes and to express the potency of BoNTs defined in lethal dose—LD50 Units (which corresponds to the quantity of toxin necessary to kill 50% of injected mice). The MBA also offers a high level of sensitivity, with limits of detection as low as 5–10 pg/mL [88,89,90].

Despite these advantages, several issues can be raised concerning the “3 R’s rule”: Reduce, Refine, Replace. First ethical considerations for the well-being of the animals have been raised due to the large number of mice needed for such assays [91,92]. The accuracy of the MBA is also questionable with an error rate estimated up to 40% due to a high variability between laboratories and investigators who need to be qualified and trained [93,94]. In addition, this assay is very time-consuming as test animals must be monitored up to 96 h. It is estimated that more than 40 animals per patient need to be used [95]. Finally, interspecies differences between mice and humans may lead to misinterpretations of the results. This is the case with BoNT/B where the potency is higher in mice than in humans, owing to a residue difference in the SYT-II receptor resulting in a lower binding affinity of BoNT/B for human SYT-II [96,97]. Similarly, BoNT/C and BoNT/D are differentially active between animals and humans [98,99].

In order to overcome these limitations, efforts have been made to develop alternative solutions. For example, the nonlethal mouse Digit Abduction Score (DAS), recently adapted to rats, has been developed to detect and identify BoNTs serotype after intramuscular injection into lower limb skeletal muscle [100,101]. Nonetheless, these assays still require large numbers of animals and lack precision and reproducibility due to subjective read-out characterized by the retraction of the limb.

#### 3.1.2. Ex Vivo Assays

Other approaches to test BoNTs based on ex vivo models have been explored. They consist in a continuous measurement of the twitch force elicited by electric stimulation of an isolated explant of adult rodent muscles and nerves in a physiological bath. The mouse or rat phrenic nerve hemidiaphragm (PNHD) assay uses isolated hemidiaphragm muscle with the attached phrenic nerve from euthanized rodents [69,102,103]. More recently, a new ex vivo assay was developed in order to evaluate the effect of different subtypes of BoNTs on smooth muscles issues from bladder preparations [104].

Altogether, ex vivo assays prove particularly useful to decipher BoNTs mechanism of action and identify intracellular events and receptors involved in neurotoxicity. Although ex vivo assays can replace the MBA, they are still based on the use of animals, and they require the implementation of expensive technical platforms necessary for all the measurements.

### 3.2. Cell-Based Assays

In the last decade, significant progress in the assessment of BoNTs’ potency has been made toward reducing animal use. Different cell-based in vitro assays (CBA) using specific neuronal cell lines have been implemented to detect multiple steps of BoNTs activity including membrane receptor binding, toxin uptake, translocation, and intracellular substrate cleavage [92,105]. These CBAs are currently the best alternative and the most sensitive assays for studying BoNTs potency and activity.

One of the main read-outs used in CBA is SNAP25 cleavage which can be quantified either by Western Blotting [106,107], ELISA [108,109], or immunofluorescence assays [110,111,112]. Due to the promising results obtained with these systems, especially regarding the determination of pharmacokinetics of purified BoNTs, these assays are now considered as a gold standard in translational study BoNTs.

Furthermore, in the field of BoNTs research and development, for several years commercial batches of toxins have been validated with cell-based assays (BOTOX^®^ in 2011, XEOMIN^®^ in 2015, DYSPORT^®^ in 2018), which supports the relevance of such assays in this field.

#### 3.2.1. CBAs Based on Immortalized Cell Lines

Immortalized cell lines derived from animal or human cancer cells have been widely used in the field of BoNTs studies for several reasons: (i) these cells self-renew in culture without limitations and can thus be produced in large quantities, (ii) they are easy-to-use, and (iii) relatively inexpensive [105,113,114,115]. Neuro-2a cells (mouse neuroblastoma) [116], SH-SY5Y (human neuroblastoma) [117], and SiMa cells (human neuroblastoma) [118] are now considered as efficient models for studying the biological activity of BoNTs. However, concerns about the use of immortalized cell lines have also been raised: (i) their genetic background with regard to transcriptomic and epigenetic context is different from primary neurons (such as motor neurons), (ii) the sensitivity to BoNTs can be low and varies between cell lines requiring either higher doses of toxin or longer times of exposure (2–3 days) for detectable effects [92]. Differences in sensitivity has been shown to result from factors such as variations in expression of receptors, substrates, and other cellular proteins, between different immortalized cell lines, and also the protocols used to maintain them in vitro [105]. For example, pre-incubation of cells with gangliosides increased sensitivity to BoNTs [119]. In the same way, sensitivity was increased when cells were maintained in a medium containing high concentrations of potassium or calcium. Furthermore, immortalized cells in culture fail to recapitulate many of the neurotypic properties seen in mature neurons in vivo, such as formation of functional synaptic transmission networks, making them less reliable as translational assay systems.

#### 3.2.2. CBAs Based on Neuronal Primary Cultures

Primary neurons can be obtained from different embryonic neural tissues, including spinal cord [120], dorsal root ganglion [121,122], hippocampus [123], and cortex [124], and from different species ranging from mouse to chicken embryos. Dissected primary neurons from embryos harbor a high sensitivity to BoNTs intoxication compared to immortalized cell lines [125,126,127]. Indeed, in many studies 24 h of toxin exposure was sufficient to readily measure full dose–response curves for SNAP25 cleavage (assessed by Western Blotting) compared to 48 h needed for similar assays in immortalized cells [121]. Primary neurons can be maintained for long periods (from weeks to months) of time to study BoNTs effects in a serum-free medium optimized for neurons preventing the proliferation of non-neuronal cells [128]. Unfortunately, the use of primary neurons often requires the sacrifice of pregnant animals to obtain enough dissected embryos to set up primary neuron cultures for CBA. Another limitation is the heterogeneous nature of primary cultures derived from different dissection leading to variable neuron-glia mix cultures and ultimately variation in measurement of toxin potency. Finally, CBA based on primary culture suffer from biases mediated by species-specificity of BoNTs activity similarly to MBA and other assay based on animal biological resource. Differences related to the species are likely associated with the fact that BoNT receptors and/or SNARE proteins differ from one species to another [129].

## 4. Emerging Cell-Based Assays Using Human iPSCs Derivatives

### 4.1. Human Induced Pluripotent Stem Cells

Since their discovery more than 20 years ago, hPSCs have ushered in a new era for the fields of stem cell biology and regenerative medicine, as well as disease modeling and drug discovery.

The “physiological” and “natural” source of human pluripotent stem cells are the human embryonic stem cells (hESCs), which are derived from human blastocyst of in vitro fertilized embryo [130]. While hESCs are considered as the gold standard of hPSCs, their use has raised ethical issues in different countries. Induced pluripotent stem cells (iPSCs), first described by S. Yamanaka and his group with mouse cells in 2006 then in human cells in 2007, overcome the ethical controversy associated with ESCs and represented a major breakthrough in stem cell research. The possibility of reprogramming human somatic cells into a pluripotent embryonic stem cell state through the expression of a combination of transcription factors (Oct3/4, Sox2, Klf4, and c-Myc) preserving the embryonic stage earned S. Yamanaka the Nobel Prize in Physiology or Medicine in 2012 for this breakthrough. IPSCs reprogramming methods (integrative and non-integrative) have evolved in recent years [131], as well as the tools and methods for quality control of pluripotency and assessment of genomic instabilities of cells [132,133,134]. In the same way, the ability to manage the differentiation protocols that produce neuronal, glial, muscular, and other derivatives in an increasingly reproductive way, makes these cells an exciting tool for clinical applications.

The main conditions for hPSC-based therapeutic development rely on the “holy grail” to efficiently and robustly differentiate hPSCs into all cell types of interest. Much effort has focused on the differentiation of hiPSCs into mature cell types and the last decades have witnessed the development of more and more robust and efficient protocols allowing the conversion of hiPSCs into a large panel of different cell types such as pancreatic beta cells, hematopoietic cells, cardiomyocytes, skeletal muscle cells and neural cells. Regarding the latter, relevant protocols have been developed raising the possibility to trigger neural induction [135] and next to have access to a large spectrum of neuronal subtypes such as cortical [136,137,138], dopaminergic [139,140,141], striatal GABAergic [142,143,144], hippocampal [145,146], and motor neurons [147,148].

The derivation of hPSCs regardless their origin sparked enthusiasm for the development of new models of human disease, enhanced platforms for drug discovery and more widespread use of cell-based therapy. Several pioneering studies have validated the potential of disease-specific hESCs to reflect diseases intrinsic to the cellular level [149,150,151,152]. Given that the isolation of primary diseased ESCs lines was limited by the requirement for a preimplantation genetic diagnosis in the context of in vitro fertilization, genetic manipulation of normal ESCs lines was used to introduce disease-relevant molecular defects using contemporary technologies such as RNA interference or homologous recombination [153,154,155].

### 4.2. Human iPSCs for BoNT Research and Development

In order to exploit the therapeutic potential of BoNTs, appropriate in vitro models are needed to extend the knowledge about mechanism of action, to characterize and compare serotypes, and support translation to the clinic.

There are currently few studies in the literature focusing on the use of hiPSCs for the study of BoNTs (Table 3). Although all these studies diverged in their protocols, they all described that hiPSC-derived neurons express all the necessary receptors and substrates for BoNT intoxication and that they consequently represent a highly sensitive platform for BoNT potency determination as measured by SNARE protein target cleavage. However, only two studies have shown the potential of hiPSC-derived neurons to measure at the functional level the impact of BoNT treatment [156,157].

The use of iPSCs of human origin should increase species-specific relevance and offer high sensitivity together with the possibility to compare BoNT serotypes [90]. Robust and well-characterized protocols to differentiate hPSCs into neurons are constantly in development. The methodology of neuronal differentiation has been well established in hESCs and can be directly applied to hiPSCs. Indeed, the addition of morphogens during the early neural induction steps is often used to drive differentiation towards specific neural subtypes (motor neurons, cortical neurons, dopaminergic neurons, etc.) [158,159]. These specific protocols produce populations of post-mitotic neurons capable of establishing functional synaptic networks, making functional studies and high throughput possible. In this context, the progress in hiPSCs differentiation into functionally networked neurons is revolutionizing the use of neuronal models for BoNTs research.

Many studies confirmed that iPSCs-derived neurons (i.e., motor neurons, cortical neurons, dopaminergic neurons, and sensory neurons) express all the necessary receptors and substrates for BoNT intoxication, and are sensitive for detection of several BoNTs serotypes with different potencies, making iPSC-derivatives cells of choice for BoNT research [96,157,160,161,162,163]. Motor neurons derived from hiPSCs have proved to be highly sensitive to BoNT intoxication [157,160,161]. Neurons derived from hiPSCs exhibited appropriate morphology, electrical behaviors, transsynaptic signaling and network activity. This combination of physiological relevance and neuromimetic responses enhances the relevance of the use of hiPSC-derivatives for neurotoxicology studies [164]. The appearance of network activity suggests that differentiation of hiPSCs into neuronal populations can generate synaptically mature neuronal networks. Thus hiPSC-derivatives can be incorporated into interesting functional systems to study the different therapeutic fields of application of BoNTs, for example through coculture systems using hiPSC-derived motor neurons and muscle cells to build an NMJ model [157], or using hiPSC-derived cortical neurons and astrocytes to build a cortical network model [165]. In addition, the use of pertinent coculture systems in the field of BoNT research offers the possibility to test and compare different BoNT serotypes whatever their SNARE substrate in more physio-relevant systems, also to study BoNT intoxication at functional synapses that model physiological cell–cell interactions. The development of hiPSC-based systems and the progress made in recent years on increasingly robust hiPSCs differentiation protocols has resulted in the creation of more physiological and functional models for drug screening in miniaturized format to build attractive systems that are translational for preclinic and clinic research. Combined with the progress made in high-throughput technologies, it is possible to evaluate more complex and precise functional parameters including the flow of Ca^2+^ within and between connected cells, the electrical activity, the activity of part of the system, the connection between the cells composing the system, and the activation or inhibition of a part of the system.

## 5. Challenges and Future Perspectives

The development of hiPSC-based systems and the progress made in recent years on increasingly robust hiPSCs differentiation protocols has resulted in the creation of more physiological and functional models for drug screening in miniaturized format to build attractive systems that are translational for preclinic and clinic research. Combined with the progress made in high-throughput technologies, it is possible to evaluate more complex and precise functional parameters that should open new perspectives in the field of BoNTs. In this section, we will evaluate these perspectives notably in terms of functional analyses as well as the challenges that still need to be overcome.

### 5.1. Functional Studies

Properties of ion channel function are routinely measured by patch clamp which is considered the gold standard for analysis of membrane electrical activity. Considerable effort to automate this analysis has been made for pharmacological testing of compounds to evaluate the potency of compounds-channel interactions. Progress has been achieved on the miniaturization and automation of the electrophysiological set up to increase the throughput and sensitivity of simultaneous recording of single cell, and assess cumulative or multiple compound additions [175,176]. A method combining both fluorescence-based technologies and automated patch clamp became commonly used for ion-channel-targeted drug discovery, allowing also to differentiate more easily individual sub-types of ion channels.

In the area of BoNT research, channel-specific toxins have multiple mechanisms of action, including interference of ion-channel opening. BoNT binding alters normal conformational changes required to open or close activated channels which can cause uncontrolled neuronal excitation, increased or reduced neurotransmitter release, and muscle spasm. BoNTs offer an example of toxin-induced structural change causing alterations in neuronal network electrophysiology by reducing or interrupting synaptic signals [177]. Several studies based on in vitro neuronal cultures derived from rodent stem cells demonstrated that synaptic transmission, measured through synaptic currents using patch-clamp electrophysiology, was impaired to near-total silencing in response to BoNT/A intoxication [107,178,179,180]. These studies also highlighted a synapse sub-population with specific latency toward BoNTs intoxication. Measurements of synaptic activity, revealed that glutamatergic synapses are intoxicated less rapidly than GABAergic synapses [181]. Whereas these studies on rodent stem cells provide exciting opportunities, no demonstration has been made so far of functional measurements of BoNT intoxication in synaptically active neuronal cultures derived from hPSCs. This may be partially related to the fact that generating electrophysiologically mature neuronal networks from hPSCs is still long, tedious and overly challenging.

To respond to these challenges, one possible alternative may reside in the use of multi-well microelectrode arrays (MEA). Whereas patch clamp electrophysiology records the action potential electrical activity from the intracellular space of a single neuron. MEA is a noninvasive electrophysiology technique that records the field potential electrical activity from the extracellular space of a population of neurons.

In the context of BoNT research, the combination of MEA systems and stem cells has recently been successfully applied to evaluate potency and efficiency of BoNT/A on neuronal cultures derived from mouse ESCs [182]. In this study, the use of mESCs-derived neuronal cultures, containing glutamatergic, GABAergic neurons and astrocytes, allowed assessment of alterations in synaptic transmission through burst activity measurement after BoNT/A treatment, in a dose and time dependent manner [182]. MEA measurements can also provide important functional information to establish personalized drug treatments opening a new avenue for new BoNT therapeutics. As an example, in the specific field of epilepsy, a recent study provided evidence for convulsion toxicity of new drugs and the neurological effects of antiepileptic drugs [183]. In this study, MEAs were used to evaluate antiepileptic drug responses in hiPSC-derived neurons treated with seizure-inducing drugs. The authors measured synchronized bursts firing in the epileptiform activities and compared their results to in vivo convulsive responses of the same drugs. This study also demonstrated the importance of co-culturing neurons and astrocytes to enhance the signal recorded and increase the physiological relevance of the cell-based model used.

In parallel to these electrophysiological approaches, the recent development of optogenetic techniques has generated considerable excitement in neuroscience research. Optogenetics, first developed in 2005, is a non-invasive genetic method that involves the use of light to control the activity of cells with both high temporal and high spatial precision [184]. As optogenetics allows selective targeting of individual cell types and activation or inhibition of their activity with a millisecond-scale time resolution, it offers a high degree of specificity and control over cellular network activity. This method requires the use of light-activated proteins called microbial opsins such as channelrhodopsin-2 (ChR2) for activation and halorhodopsin (NpHR) for silencing of cells [185,186]. ChR2 is a light-gated cation channel activated by blue light (470 nm) with high temporal precision. It is used to depolarize neurons and thus generate action potentials. Conversely, NpHR is a light-gated chloride pump activated by yellow light (580 nm) causing hyperpolarization of cells used to silence excitable cells. In neurology, optogenetics has enabled scientists to causally link cellular circuits, behavior, and function.

Recently, BoNTs have been the subject of optogenetic engineering. Liu and colleagues described a blue light activatable BoNT engineered to disrupt excitatory neurotransmission resulting in persistent synaptic inhibition triggered by light [187]. Using this optogenetic approach, the long-term disruption of synaptic transmission induced by the direct application of BoNTs and the resulting changes (in terms of behavior, cellular mechanisms, ions transient) could be evaluated after activation of photoactivatable-engineered synapses, which will allow the entire system to be controlled. In that way, the effects of different BoNTs serotypes could be measured after overactivation of the synapse with optogenetics, in order to establish guidelines for future therapeutics.

### 5.2. Towards More Relevant hiPSC-Based Models for BoNT Research

Although hiPSC-derived systems are relevant tools with considerable advantages for studying BoNTs, including for high-throughput analyses, several concerns can still be raised on the relevance of these systems. First, the developmental stage of hiPSC-derived neurons remains unclear. In general, in vitro disease modeling techniques use short-term cultures that fall short of producing really mature hiPSC-derived neurons. To overcome this limitation, approaches such as prolonged neural differentiation [188], long-term cultures [189,190], or co-culture with astrocytes [191], have proven enhanced neuronal maturation. More recently, tri-cultures of hiPSC-derived neurons, astrocytes and microglia have also been described allowing the study of neuron-glia interactions [192]. This is of particular interest for studying and developing BoNT as there is some evidence suggesting that BoNT can also exert effects on glial cells, including astrocytes and microglia [193,194,195]. The establishment of such mixed cocultures should facilitate the evaluation of new aspects of BoNT biology.

Another important concern resides in the fact that BoNT intoxication in humans arises in the context of complex multicellular tissues and organ systems and is dependent on multiple interactions occurring between cells, extracellular matrices, and pathogens. Consequently, more complex iPSCs-based differentiation systems (i.e., 3D systems, engineered tissue, organ-on-chip, and organoids) are currently being developed to faithfully recapitulate human tissue-level and organ-level dysfunctions [196,197,198]. Organoids are unique, in that they are self-organizing, 3D culture systems that are highly similar to human organs [199]. A variety of iPSCs-derived organoids mimicking the brain [200], retina [201], liver [202], lung [203], kidney [204], and heart [205] have been developed. Organoids exhibit sophisticate 3D architecture and contain many of the cell types found in the in vivo tissues they reproduce. In this part we will mainly discuss the progress made on the modeling of the NMJ, the target of choice for the study of BoNTs.

Significant advance in the generation of 2D culture systems and recently in 3D culture systems open new perspectives for BoNTs research [206]. While 2D cultures are limited because they do not reproduce the entire structure and hierarchical connectivity that is observed in 3D cultures, and do not mimic the native tissue structure, they remain advantageous in terms of quality, reproducibility, and timeliness (Table 4). The expansion of spheroids and organoids systems provide models with enhanced structural and morphological relevance in the generation of models that resemble a human NMJ system.

Faustino Martins and colleagues established a 3D human neuromuscular organoid system that self-organized and formed functional NMJ [207]. Muscle cell functionality was measured with the calcium imaging combined to MEA to evaluate neuronal network activity. The main advantages of this model are that such a culture can be maintained for longer periods (months to years), and that they allow the generation of unique structures in which all the components of the NMJ are present including the Schwann cells, which are essential for the NMJ maturation and homeostasis. This model reveals physiological endogenous interactions between all cell types inside organoids, giving access to the study of key human developmental events of the NMJ and allowing the contribution of each cell type to NMJ genetic disorders to be deciphered. Another study from Andersen and colleagues established 3D human cortico-motor assembloids to recreate a multi-synaptic circuit which regulates the neuronal activity in the hindbrain and spinal cord to generate coordinated movement [208]. The functionality of the synaptically connected assembloids was measured using calcium imaging and patch clamp recordings. Glutamate and optogenetic stimulations used in this study revealed a contractile response of 3D muscle to neuronal stimulation. This system also illustrates the ability of 3D cultures to self-assemble to form functional circuits that can be used to understand development and disease. iPSC-derived 3D human neuromuscular organoid or assembloid systems constitute uniquely attractive system to study neuromuscular disease and to develop or validate candidate therapies.

The 3D models provide important information in understanding several mechanisms underlying neuronal network function and homeostasis, but they do not fully replicate the exact human physiological functional units formed by the connection of different excitable cell types, resulting from separated cellular microenvironments. To address this need, compartmentalized microfluidic technology represents an alternative to 3D system to recreate NMJ physiology of human in vitro. Bellman and coworkers, as well as Osaki and colleagues, developed a microfluidic systems to model NMJ by creating a specific cell microenvironment for hPSC-derived motor neurons and hPSC-derived muscle cells [209,210]. Such a microfluidic system facilitates imaging and quantitative functional evaluation. These authors generated a functional NMJ that matured in few weeks on microfluidic chips and used optogenetics to demonstrate the functionality of neuromuscular circuits. With this system, pathological behaviors associated with neuromuscular diseases (e.g., degeneration and death of motor neurons, muscle atrophy) can be recapitulated using patient-derived iPSCs to generate motor neurons and muscle cells used. Overall, the throughput allowed by 3D or microfluidic set-up remains until now limited to only medium throughput applications in the drug discovery pipeline.

Despite the fact that synapses including the NMJ are complex multi-cellular 3D structures, the majority of drug screening and safety or efficacy measurement are still carried out using mono-culture of cells grown a 2D monolayer [211,212]. Not surprisingly, drug responses in these in vitro models are often poorly predictive of in vivo situations. In the field of BoNTs study, the generation of robust 3D and multicellular systems would make it possible to compare all aspects of the different BoNTs serotypes and also to focus on new pathways that could be interesting for the development of future BoNT-based therapeutics.

## 6. Conclusions

Recent studies have established the versatility of hPSCs to generate clinically and biologically relevant multicellular models. For BoNT detection and mechanistic studies, various cell models currently exist, but none examine BoNT function with human-specific relevance while exhibiting high sensitivity.

To study BoNTs, preferential models have been developed over the years: in vivo, BoNT-induced muscle relaxation is assessed in mice and rats (MBA); BoNT potency was evaluated ex vivo on mice hemidiaphragms, measuring muscle contractility; and BoNT potency was preferentially assessed in vitro in rodent primary cells (CBA). All these models have several limitations such as the requirement for significant technical expertise and highly skilled personnel to implement the assays, reproducibility and stability, cross species differences limiting translation of data from animals; they also raise important ethical concerns regarding the large number of rodents require by these assays. In this context, more physiologically and predictively relevant models are needed. Human iPSCs technology offers the opportunity to study BoNTs in relevant humanized context making cell-based assays crucial for the advancement of BoNT research and drug discovery. The continuous development of robust protocols for the differentiation of hiPSCs into various neuronal sub-populations opens the door to comparative studies of different BoNTs serotypes in different neuronal sub-populations. Several studies have shown that hiPSC-derived neurons were highly BoNT-sensitive models with different sensitivities depending on the BoNTs serotypes. Moreover, the generation of a variety of neurons of different phenotypes (motor neurons, GABAergic, glutamatergic, dopaminergic, sensory, etc.) allows the development of more complex, translational, and predictive cell-based systems to help in the development of a panel of BoNT-based future therapeutics in the areas of movement disorders, neurological diseases, neurodegenerative diseases, pain, and other disorders.

The complexity, in terms of the number and interconnectivity of cell types, maturity, and differentiated state of most cells of the human nervous system is difficult to replicate in cell-based models. One important aspect of cell-based assay is the read-outs often provide information on cell viability, morphology, and neuron-specific functional activity in addition to the primary output. Progress in high-throughput assays can provide information on multiple functional parameters such as electrical activity, network synchrony, calcium transient. In this sense, human neuronal cells derived from iPSCs are attractive models because they exhibit the function, connectivity, and behavior of mature neurons in a synaptic network.

To identify network activity in a high-content manner, multiple laboratories have employed electrophysiology, MEA, and more recently optogenetics platforms to evaluate in a high-content context the activity of a neuronal network by using relevant read-outs such as calcium signaling. Several studies showed the efficacy of these techniques to record the activity of synaptic networks which is an emerging target for the development of future therapeutics. Taking advantage of hiPSCs-derived systems, sophistication of models (3D, organoids), and high-throughput functional assays, the study of the effect of different BoNTs serotypes on synapses such as NMJ, but also on other neuronal synapses, should open the way to new BoNT-based therapeutics.

## Figures and Tables

**Figure 1 ijms-22-07524-f001:**
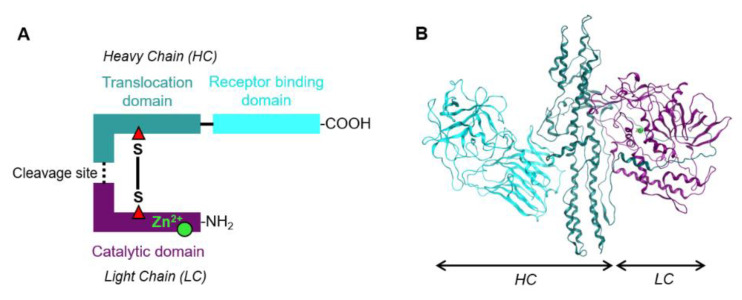
BoNT structure. (**A**) Schematic illustration of BoNT/A structure composed of the light chain (LC) peptide fragment (50 KDa) linked with a disulfide bond (S-S) to the heavy chain (HC) peptide fragment (100 KDa). The LC is a zinc (Zn^2+^) endopeptidase responsible for catalytic activity, and the HC comprises a receptor binding domain responsible for neurospecific targeting and a translocation domain responsible for the LC translocation. (**B**) Illustration of BoNT/A crystalline structure. Same colors represent same elements in both parts of the figure.

**Figure 2 ijms-22-07524-f002:**
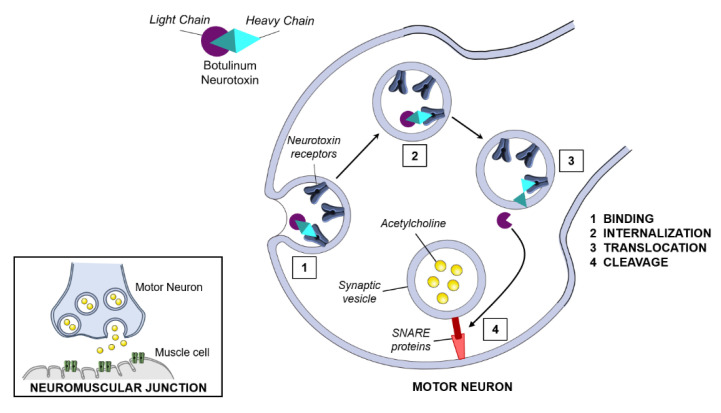
Schematic representation of BoNTs cellular mechanism of action. BoNTs intoxication consists of four steps: binding, internalization, translocation, and cleavage. BoNTs bind to specific receptors via its heavy chain, leading the internalization into the neuron. Once internalized into an endosome, the decrease in pH activates the heavy chain translocation domain, which detaches from the light chain. The light chain moves into the cytosol and binds its appropriate SNARE substrate based on BoNTs serotype. SNARE cleavage blocks the release of acetylcholine into the neuromuscular junction affecting motor neuron function and the resulting muscle contraction.

**Table 1 ijms-22-07524-t001:** BoNTs serotypes characteristics.

Serotype	BoNTs Receptor	Reference	Target	Reference
A	SV2A, SV2B, SV2C	[40,41]	SNAP25	[42,43,44,45]
B	SYT-I, SYT-II	[46,47,48]	VAMP	[49]
C	GD1b, GT1b	[50]	SNAP25, Syntaxin	[42,51,52]
D	SV2A, SV2B, SV2C	[53]	VAMP	[43,49,54]
E	SV2A, SV2B	[55,56]	SNAP25	[43,44,45]
F	SV2A, SV2B, SV2C	[57]	VAMP	[43,49,54]
G	SYT-I, SYT-II	[47,48,58,59,60]	VAMP	[61]
H	SV2A, SV2B, SV2C	[62]	VAMP	[63,64]
X	not identified		VAMP1/2/3/4/5, Ykt6	[33,34]

BoNTs: botulinum neurotoxins; GD/GT: gangliosides; SNAP25: synaptosomal-associated protein 25 KDa; SV2: synaptic vesicle protein 2; SYT: synaptotagmin; VAMP: vesicle-associated membrane protein.

**Table 2 ijms-22-07524-t002:** BoNTs administration authorities-approved therapeutic indications.

Commercial Name	Formulation	Serotype	Approved Indications	Date
DYSPORT^®^	Abobotulinumtoxin	A	Blepharospasm	1990
			Hemifacial spasm	1990
			Cervical dystonia	2009
			Glabellar and lateral canthal lines	2009
			Adult upper limb spasticity	2015
			Pediatric lower limb spasticity	2016
			Adult lower limb spasticity	2017
			Pediatric upper limb spasticity	2019
BOTOX^®^	Onabotulinumtoxin	A	Blepharospasm	1989
			Strabismus	1989
			Cervical dystonia	2000
			Glabellar lines	2002
			Hyperhidrosis	2004
			Adult upper limb spasticity	2010
			Chronic migraine	2010
			Neurogenic overactive bladder	2011
			Urinary incontinence	2011
			Idiopathic overactive bladder	2013
			Adult lower limb spasticity	2016
			Pediatric upper/lower limb spasticity	2019
			Pediatric spasticity	2020
XEOMIN^®^	Incobotulinumtoxin	A	Blepharospasm	2010
			Strabismus	2010
			Cervical dystonia	2010
			Glabellar lines	2011
			Upper limb spasticity	2015
			Sialorrhea/Excessive drooling	2018
			Blepharospasm/Involuntary blinking	2019
			Chronic sialorrhea	2020
MYOBLOC/NEUROBLOC^®^	Rimabotulinumtoxin	B	Blepharospasm	2010
			Strabismus	2010
			Cervical dystonia	2010
			Glabellar lines	2011
			Chronic sialorrhea	2019

**Table 3 ijms-22-07524-t003:** Use of hiPSCs for BoNT studies.

hiPSC-d Neuronal Type	Source	Culture Time	BoNT Serotypes Tested	Treatment Duration	Assay/Read-Out	Outcomes	Reference
GABA	Prop	4, 7 days	A, B, C and E	6, 16, 24, 48 h	WB/SNARE-cleavage	EC50	[163]
GABA	Prop	2, 14 days	A and catalytically inactive A	48 h	Transcriptomic analysis	Transcriptomic signature	[166]
GABA	Prop	7 days	A, B, D, E and F	48 h	WB/SNARE-cleavage	EC50	[167]
GABA	Prop	7 days	A	48 h	WB/SNARE-cleavage	EC50	[168]
NSC	Prop	28 days	A	48 h	WB/SNARE-cleavage	EC50	[168]
GABA	Prop	7 days	A, B and D	48 h	WB/SNARE-cleavage	EC50	[169]
NSC	Prop	28 days	A, B and D	48 h	WB/SNARE-cleavage	EC50	[169]
GABA	Prop	7 days	A	48 h	WB and ELISA/SNARE-cleavage	EC50	[170]
NSC	Prop	28 days	A	48 h	WB and ELISA/SNARE-cleavage	EC50	[170]
GABA	Prop	7 days	A	6 h	WB/SNARE-cleavage	EC50	[171]
GABA	Prop	7 days	FA, F subtypes and B	48 h	WB/SNARE-cleavage	EC50	[172]
GABA	Prop	7 days	A subtypes	48 h	WB/SNARE-cleavage	EC50	[173]
GABA, Gluta, MN, Peripheric	Prop	14 days	A and E	24 h	WB/SNARE-cleavage	EC50	[160]
MN	Prop	14 days	A and E	24 h	WB/SNARE-cleavage	EC50	[174]
GABA	Prop	14 days	A, B and modified B	24 h	NT Release assay/Glycine concentration	EC50	[96]
GABA, Gluta, MN, Dopa	Prop	14 days	A, B, C, E and F	48 h	WB/SNARE-cleavage	EC50	[161]
MN/Coculture with human myotubes	Pub	>25 days	A	<1 h	NMJ model in chips/Frequency of myotubes contraction	IC50	[156]
MN	Pub	>28 days	A	48 h	WB/SNARE-cleavage	IC50	[162]
GABA, Gluta, MN, Peripheric	Prop	14 days	A and E	24 h	WB/SNARE-cleavage	EC50	[157]
MN/Coculture with human myotubes	Prop	14 days	A and catalytically inactive A	6, 24 h	NMJ model P96/Frequency of myotubes contraction	3 doses	[157]

hiPSC-d: human induced pluripotent stem cell-derived; Dopa: dopaminergic neurons; GABA: GABAergic neurons; Gluta: glutamatergic neurons; MN: motor neurons; NMJ: neuromuscular junction; NSC: neural stem cells; Prop: property of Cellular Dynamics International, NCardia or Lonza depending on the study; Pub: published protocol; EC50: half maximal effective concentration; IC50: half maximal inhibitory concentration; NT: neurotransmitter; WB: Western blot.

**Table 4 ijms-22-07524-t004:** Comparison of 2D and 3D cell culture strategies.

	2D Culture	3D Culture
Culture quality	High reproducibility, long-term culture, simplicity of culture	Low reproducibility, cultures more difficult to carry out
In vivo modeling	Not mimic the native structure of the tissue	Tissues and organs are in 3D form
Extracellular environment	No in vivo-like microenvironment and no niches	Environmental niches are created
Cell characteristics	Loss of diverse phenotype and polarity	Preservation of diverse phenotype and polarity
Access to essential compounds	Unlimited access to oxygen, nutrients, signaling molecules	Variable access to oxygen, nutrients, signaling molecules
Cost	Cheap, commercially available medium and products	Expensive, more time-consuming, fewer commercially available products

## Data Availability

Not applicable.

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
