# Peer review of "Emerging Opportunities in Human Pluripotent Stem-Cells Based Assays to Explore the Diversity of Botulinum Neurotoxins as Future Therapeutics"

_ijms, 2021, doi:10.3390/ijms22147524_

Round 1
Reviewer 1 Report
This is a well-written review on using human pluripotent stem-cells based assays to explore the diversity of botulinum neurotoxins as therapeutics. However, as authors focusing on using hiPSC based assays for future therapeutic applications of BoNT, it would be helpful to give an overall of the current status of using hiPSC as assays for BoNT in the pharmaceutical industry. While authors listed the challenges of hiPSC based assays, they may elaborate on the particular challenges facing pharmaceutical development, especially given that the high specificity, reproducibility, and precision needed for assays used for pharmaceutical applications, and how to overcome those challenges.
Reviewer 2 Report
The manuscript provides an overview of detection methods for botulinum neurotoxins with an emphasis on human pluripotent stem cells. The review starts very nicely but when the authors get to human pluripotent stem-cells literature the manuscript goes off rails and the reader becomes lost on what is the review is about. The language, the cited literature and the structure of the manuscript are mainly appropriate however their sudden discussion of neurodegenerative and other neurological disorders distracts from the title and the main point of the review.
Text 358-406 doesn’t add anything to the title ‘Emerging opportunities in human pluripotent stem-cells based assays to explore the diversity of botulinum neurotoxins as future therapeutics’, otherwise the authors need to add 'disease focus' in the title.
The most relevant chapter ‘Human iPSCs for BoNT research and development’ lacks relevant details and leaves the reader empty-handed. There are no specific examples for cells used, their sources, timeline-inclusive protocols, what was measured for different serotypes et cetera. It looks more like a newspaper article rather than a useful scientific piece of work which could be used without going into references. There is no critical look at iPSCs whatsoever, it would be useful to see the time involved in getting these cells ready for testing and resources required, costs, et cetera.
Further chapters are also missing on specific results obtained, sensitivities for different serotypes.
476 Several studies based on in vitro neuronal cultures derived from rodent stem cells demonstrated that synaptic transmission, measured through synaptic currents using patch-clamp electrophysiology, was impaired to near-total silencing in response to BoNT/A intoxication [181]. These studies also highlighted a synapse sub-population with 479 specific latency toward BoNTs intoxication. – Do you need to add more references to justify ‘Several’ and ‘These’.
The last chapter on Optogenetics again slips into the neurodegeneration field and leaves the reader without insight into opportunities in human pluripotent stem-cells based assays to explore the diversity of botulinum neurotoxins. Can their outline methodology addressing this topic with quantitative outcomes?
Overall, it would be good to see more quantitative parameters of different approaches to measure botulinum activities.
Minor points:
318 hPSCs have ushered in an exciting new era – Consider removing ‘exciting’ to be more balanced.
Similarly
349 The derivation of hPSCs regardless their origin sparked widespread enthusiasm for the development of new models of human disease, enhanced platforms for drug discovery and more widespread use of cell-based therapy.
-Consider removing ‘widespread’, because enthusiasm is sufficient
351 A couple of pioneering studies have – it is too colloquial. Consider ‘Several pioneering studies’
Reviewer 3 Report
This review presents an emerging family of cell-based assays based on the newly developed human Pluripotent Stem Cells. Several studies have shown that hiPSC-derived neurons are highly BoNT-sensitive models. Interestingly, the generation of a variety of neurons of different phenotypes (motor neurons, GABAergic, glutamatergic, dopaminergic, sensory) permits the development of more complex, translational and predictive cell-based systems to help in the development of a panel of BoNT-based future therapeutics in the areas of movement disorders, neurological diseases, neurodegenerative diseases, pain and other disorders. This review is timely and valuable and has presented several benefits while keeping the limitations highlighted for ways to overcome the current challenges.
There is one question related to mix cultures of neuron glia, can it be pure glia cultures of for example astrocytes? and would this be a benefit or a limitation? One can see both aspects, as in the real life, the system functions together, not separately.
Round 2
Reviewer 1 Report
It can be accepted after some editorial corrections.
Reviewer 2 Report
The authors improved the manuscript as suggested.